# Association between microbial composition, diversity, and function of the maternal gastrointestinal microbiome with impaired glucose tolerance on the glucose challenge test

Caitlin Dreisbach[1,2]*, Stephanie Prescott[3,4], Jeanne Alhusen[2], Donald Dudley[5], Giorgio Trinchieri[3], Anna Maria Siega-Riz[6]

1 Data Science Institute, Columbia University, New York, NY, United States of America, 2 School of Nursing, University of Virginia, Charlottesville, VA, United States of America, 3 College of Nursing, University of South Florida, Tampa, FL, United States of America, 4 Center for Cancer Research, National Cancer Institute, Bethesda, MD, United States of America, 5 Division of Maternal-Fetal Medicine, University of Virginia Health System, Charlottesville, VA, United States of America, 6 School of Public Health and Health Sciences, University of Massachusetts, Amherst, MA, United States of America

* c.dreisbach@columbia.edu

**Data Availability Statement:** Metagenomic data are available from the NCBI SRA database at

## Abstract

Over the last two decades, the incidence of gestational diabetes (GDM) has almost doubled resulting in almost 9% of pregnant women diagnosed with GDM. Occurring more frequently than GDM is impaired glucose tolerance (IGT), also known as pre-diabetes, but it has been understudied during pregnancy resulting in a lack of clinical recommendations of maternal and fetal surveillance. The purpose of this retrospective, cross-sectional study was to examine the association between microbial diversity and function of the maternal microbiome with IGT while adjusting for confounding variables. We hypothesized that reduced maternal microbial diversity and increased gene abundance for insulin resistance function will be associated with IGT as defined by a value greater than 140 mg/dL on the glucose challenge test. In the examination of microbial composition between women with IGT and those with normal glucose tolerance (NGT), we found five taxa which were significantly different. Taxa higher in participants with impaired glucose tolerance were *Ruminococcacea* ($p = 0.01$), *Schaalia turicensis* ($p < 0.05$), *Oscillibacter* ($p = 0.03$), *Oscillospiraceae* ($p = 0.02$), and *Methanobrevibacter smithii* ($p = 0.04$). When we further compare participants who have IGT by their pre-gravid BMI, five taxa are significantly different between the BMI groups, *Enterobacteriaceae*, *Dialister micraerophilus*, *Campylobacter ureolyticus*, *Proteobacteria*, *Streptococcus Unclassified* (species). All four metrics including the Shannon ($p < 0.00$), Simpson ($p < 0.00$), Inverse Simpson ($p = 0.04$), and Chao1 ($p = 0.04$), showed a significant difference in alpha diversity with increased values in the impaired glucose tolerance group. Our study highlights the important gastrointestinal microbiome changes in women with IGT during pregnancy. Understanding the role of the microbiome in regulating glucose tolerance during pregnancy helps clinicians and researchers to understand the importance of IGT as a marker for adverse maternal and neonatal outcomes.

accession number PRJNA862188. Clinical data is available through the Obstetric and Neonatal Outcomes Study (ONOS) at the University of Virginia. Researchers interested in collaborating can email Amanda Urban (AJR5Y@hscmail.mcc. virginia.edu). Approval from the ONOS Data Access Committee is available for researchers who meet the criteria for access to confidential data.

**Funding:** C.D.: National Institute of Nursing Research (NINR), F31NR017821 and the Association of Women's Health, Obstetric and Neonatal Nurses (AWHONN), 2018 March of Dimes Margaret Comerford Freda "Saving Babies, Together®" Award. The funders had no role in study design, data collection and analysis, decision to publish, or preparation of the manuscript.

**Competing interests:** The authors have declared that no competing interests exist.

## Introduction

Pregnancy is a period of substantial metabolic and physiologic changes to support the growth of the placenta and fetus. One major adverse complication during pregnancy related to fetal growth and maternal metabolic function is gestational diabetes mellitus (GDM). GDM is characterized by persistent maternal hyperglycemia and insulin resistance that begins to resolve after delivery of the placenta [1]. The incidence of GDM doubled between 2000–2010, resulting in almost 9% of pregnant women diagnosed with GDM [2,3]. This increase is concurrent with rising rates of obesity, with now over 60% of women entering into pregnancy with a body mass index classified as overweight or obese [4]. Both obesity and a diagnosis of GDM increase the risk of type two diabetes mellitus later in life and increase the risk of childhood obesity for offspring [5]. Infants of diabetic women may also suffer from macrosomia or hypoglycemia after birth that may be persistent or profound requiring glucose testing and intervention [6].

Impaired glucose tolerance (IGT), sometimes referred to as pre-diabetes mellitus, has historically not been well-defined in pregnancy due to the lack of clinical research [1]. IGT is defined as a single abnormal blood glucose value on an oral glucose test [7]. A study conducted by Yang et al. found that a diagnosis of IGT increased the risk for premature rupture of membranes, preterm birth, and neonatal birthweight above the 90[th] percentile even after adjusting for confounding factors [8]. While we understand the implications of IGT outside of pregnancy populations, few studies have taken into consideration the heterogeneity and biological underpinnings of impaired glucose tolerance during pregnancy.

During normal gestation, maternal insulin secretion increases by almost 250% to maintain normal maternal blood glucose levels [5,9,10]. Increasing insulin resistance seen throughout gestation is primarily the result of placental secretion of tumor necrosis factor alpha (TNFa) and small contributions from maternal plasma cortisol and leptin [11]. Elevated TNFa levels induce insulin resistance through modifications of insulin signaling which trigger the breakdown of fat in adipose tissue [12]. This breakdown of fat leads to the secretion of free fatty acids into the maternal circulation and are transported as triglycerides to muscle [13]. Increases in TNFa during gestation and interrupted glucose transfer and insulin signaling may free up essential fatty acids and glucose for use in fetal growth. Obesity is already known to induce low-level inflammation and increased TNFa and leptin which, if chronic, results in insulin resistance [12]. The combination of obesity and the normal adaptations for fetal nutrient regulation may result in pancreatic beta cell dysfunction as seen after chronic exposure to inflammatory cytokines in type II diabetes [14]. Women with GDM do exhibit significantly higher levels of TNFa and leptin [15] compared to women without this disorder, however, little is known about how these factors are at play in impaired glucose tolerance. Understanding the biology of impaired glucose tolerance during pregnancy is critical for the health of both the mother and her infant.

The maternal gastrointestinal microbiota may play an important role in the regulation of inflammation and host blood glucose levels. The microbiota are the fungi, viruses, and bacteria that live within the gastrointestinal tract or other body surfaces (i.e., vagina, skin, oral mucosa). Obesity is associated with changes in microbial diversity during pregnancy [16] and gastrointestinal microbial ecology [17]. Perhaps more important than the microbial taxa abundance is the functional role that microbes may play in host metabolic functions. The gastrointestinal microbiota has been shown to contribute to metabolic changes during pregnancy, including inflammation, reduced insulin sensitivity, and excessive weight gain [18]. However, there are no human studies to date that have examined the association of the microbiota, in both composition and function, with impaired glucose tolerance on the glucose challenge test screening in the late second trimester of pregnancy. The purpose of this study is to examine the

association between maternal gastrointestinal microbial taxa, diversity, and function with impaired glucose tolerance during pregnancy.

## Materials and methods

### Data collection and biospecimen processing

Biospecimens assessed for inclusion in this study (n = 109) were collected by participants who gave their consent for the Obstetric and Neonatal Outcomes Study (ONOS) at the University of Virginia Medical Center. ONOS is a biobank of serum blood, fecal, placental, and cord blood biospecimens. Inclusion criteria for participants enrolled in ONOS includes, 1) a confirmed pregnancy at less than 20 weeks of gestation at consent, 2) anticipation of an uncomplicated singleton pregnancy, 3) maternal age between 18–45 years old, 4) primary language as either English or Spanish, and 5) planning to remain in the area for one year following delivery with the willingness and ability to provide consent. Exclusion criteria includes, 1) pre-existing diabetes, 2) chronic hypertension, 3) a multiple gestation pregnancy, 4) a diagnosis of fetal anomaly requiring surgery, 5) previous Rh iso-immunization that required a transfusion, or 6) a significant medical condition that requires long-term medication including, but not limited to, chronic thyroid disease, autoimmune disorders or steroid use.

Maternal fecal biospecimens were self-collected by participants using a BD BLL™ Culture-Swab around the time of the glucose challenge test (GCT) screening for GDM. After collection, biospecimen swabs were transported on a daily basis from the prenatal clinic to the processing laboratory. Biospecimens were recorded upon arrival to the laboratory and dried for three days. Biospecimen in a microcentrifuge tube were combined with a Thermofisher DNA Catch all Reagent. The end of the biospecimen swab was rotated in the centrifuge tube with the reagent a minimum of five times before removal of the swab to agitate the DNA. The biospecimen in the tube was vortexed for 15 seconds before an incubation at 60 degrees Celsius for one minute. After incubating, the biospecimen was vortexed again for 15 seconds and then heated at 98 degrees Celsius for two minutes. A final vortex for 15 seconds was completed before storage in deep freezer storage at -70-80 degrees Celsius. The University of Virginia Institutional Review Board approved all study procedures. Written or verbal consent was not required because eligible individuals in the parent study provided written consent for use of biospecimens in future research studies.

### Clinical metadata and variable definitions

All relevant clinical metadata for enrolled ONOS participants with a microbiome biospecimen (n = 109) was sourced from the EPIC electronic medical record system (Epic Systems Corporation) by a trained research assistant. Relevant clinical data were defined as the factors that may influence the gastrointestinal flora or place an individual at high risk of developing IGT or GDM. Data include, but are not limited to, maternal past medical history (e.g. diagnosis of diabetes or a metabolic disorder), medication administration (e.g., antibiotic administration), and prenatal data (e.g., gravida, gestational weight gain, blood glucose values) that were relevant to delivery sourced from the participant's medical record. No newborn medical records were accessed. De-identified clinical variables in the dataset were verified by two study team members for accuracy.

### Biospecimen plating, DNA extraction, and shotgun metagenomic sequencing

Biospecimens were obtained from the ONOS biorepository in April 2019 and transported to the National Cancer Institute (Bethesda, Maryland) for plate processing and DNA extraction.

Each biospecimen was labeled with a unique identifier corresponding to a well on a 96-well bead plate with four control wells. The automated DNA extraction was performed with the Qiagen MagAttract PowerMicrobiome DNA/RNA isolation kit (cat#27500-4-EP). Qubit quantification was used to confirm DNA concentration levels. DNA library preparation of fragmented sequences was completed using the Illumina Nextera DNA Flex Library Prep kit. Shotgun metagenomic sequencing was implemented on the Illumina NovaSeq 6000 platform at the National Cancer Institute Microbiome Sequencing Core (Frederick, Maryland).

## Quality control of biospecimens and data analysis

Paired-end FASTQ sequencing files were used as input into the Just Another Microbiology System (JAMS) pipeline (version 1.39) [19] for quality control, assembly of contigs, as well as a taxonomic and functional assignment. The JAMS pipeline included two phases, JAMSalpha and JAMSbeta. In JAMSalpha, paired-end sequencing reads were quality trimmed using Trimmomatic [20], aligned to the human genome while removing host sequences using Bowtie2 [21], and assembled into contigs and mapped to the microbial genome using MEGAHIT [22]. Reads that were unable to be mapped to contigs were classified with k-mer analysis using kraken [23]. All contigs and k-mer assigned reads were assigned to a taxonomic classification called the last known taxon (LKT). All biospecimens in this analysis had at least 55% of assembled contigs passing quality control. Contigs were annotated using Prokka (version 1.14) [24] and the predicted proteome was functionally annotated using InterProScan 5 [25]. To obtain the Gene Ontology (GO) term relative abundances, the base counts of InterProScan annotated genes attributed to a given GO term were aggregated by summing and were then divided by the total number of base pairs used for assembly for that biospecimen. Assessment of assembled contigs showed no bias between biospecimens based on maternal pre-gravid BMI, antibiotic administration, diagnosis of gestational diabetes, or a clinical value of IGT at the GCT. The JAMSalpha component of the pipeline was implemented on the National Institutes of Health high-performance computer, Biowulf. Downstream analysis was run on Rivanna, the high-performance computer at the University of Virginia.

Participant biospecimens were grouped by maternal pre-gravid BMI status as either normal weight (BMI $<24.9$ kg/m$^2$), overweight (BMI 25.0–29.9 kg/m$^2$), or obese (BMI $>30$ kg/m$^2$). Pre-gravid BMI was calculated using the EPIC documented pre-gravid weight and height. Three participants had a BMI of less than 18.5 kg/m$^2$ (range 18.20–18.43 kg/m$^2$) and were grouped with the normal weight group. Relevant clinical variables were compared between the BMI groups using either analysis of variance (ANOVA) and follow-up Tukey's HSD test to determine pairwise differences for parametric continuous variables or the Kruskal-Wallis rank sum test for non-parametric continuous variables. Further, chi-square testing and *posthoc* Fisher's exact test were used for categorical variables. Alpha diversity statistics were calculated using the vegan package (version 2.5–6) in RStudio [26]. Alpha diversity comparisons were analyzed using non-parametric Wilcoxon and Kruskal-Wallis tests depending on the number of BMI groups in the analysis. Spearman r correlations were calculated for the alpha diversity metrics and blood glucose values during the CGT and OGTT using the Hmisc package (version 4.3–1) [27]. The assessment of differences in taxa between BMI groups and those with IGT or NGT was completed using ANOVA. The comparison between those with IGT and NGT was completed using the Mann-Whitney U test. All heatmaps visualizations were created in the JAMSbeta pipeline phase using the ComplexHeatmap package (version 3.10) [28]. The significance level was set for 0.05 for all comparisons. Finally, false discovery rate was calculated to correct for multiple comparisons.

### Glucose challenge test and oral glucose tolerance testing during pregnancy

Current American College of Obstetricians and Gynecologists (ACOG) guidelines recommend a two-step approach to screen for GDM [29]. First, all women are recommended to have a GCT, a venous glucose measurement, one hour after a 50-gram glucose solution ingestion between 24–28 weeks of pregnancy. If the value is above the institution's screening threshold, then a three hour oral glucose tolerance test (OGTT) with 100g of glucose solution is recommended [29]. Institutional guidelines vary between 130, 135, or 140 mg/dL for the follow-up OGTT threshold. For the current study, the threshold for impaired glucose tolerance on the GCT was considered to be a blood glucose value above 140 mg/dL based on the institutional guidelines in which participants received their care.

### Inclusion and exclusion criteria

One hundred and nine (n = 109) microbiome biospecimens were assessed for inclusion in this study based on the ONOS study criteria. Data were analyzed in three subsets, 1) sample characteristics of all the ONOS participants who provided a microbiome biospecimen (n = 105), 2) participants who completed GCT screening for gestational diabetes (n = 103), and 3) participants who were determined to have IGT and underwent the OGTT (n = 16). For the first subset, the inclusion criteria were that participants had to 1) meet the ONOS inclusion and exclusion criteria, 2) enroll in ONOS, and 3) provide a microbiome biospecimen that met quality control metrics for sequencing. Four participants were excluded at this stage because their microbiome biospecimen did not contain high enough levels of DNA for extraction. For the second subset, participants were included if they completed the GCT screening during their pregnancy and there were no further exclusion criteria. Two participants were excluded at this stage (n = 1 was lost to follow-up, n = 1 had a new diagnosis of type 2 diabetes and did not complete the screening). Finally, inclusion for the third subset was that the participant had to have a greater than 140 mg/dL on the GCT with no further exclusion criteria. A STrengthening the Reporting of OBservational studies in Epidemiology (STROBE) Patient Flow diagram is available in the supplementary file.

## Results

### Participant characteristics

Participant characteristics including age, pre-gravid BMI, antibiotic administration between the time of confirmed pregnancy and collection of the fecal biospecimen, and blood glucose results on the GCT and OGTT are shown in Table 1. Women in the obese group had an average of four pregnancies compared to an average of two pregnancies in the other groups ($p<0.00$). In terms of EPIC documented race/ethnicity, 55% (n = 58) of participants were identified as Caucasian, 27.6% (n = 29) were identified as Hispanic/Latina, 11.4% (n = 12) were identified as African American, with the remaining identified as either Asian (n = 1, 0.9%), multiple races (n = 1, 0.9%), or other (n = 4, 3.8%). Further, there was a significant difference in insurance types between BMI groups ($p = 0.01$). Posthoc testing showed that the obese BMI category had a higher proportion of individuals who were uninsured and a lower proportion of individuals with private insurance to the normal weight group ($p<0.00$).

One hundred and three participants completed the CGT test between 24–28 weeks' gestation. Sixteen participants (15.2%) completed the three-hour OGTT after surpassing the threshold of 140mg/dl during the GCT. Fasting blood glucose levels were found to be significantly elevated in obese women ($p<0.00$). Four participants (3.8%) were diagnosed with GDM (n = 2 with an obese BMI, n = 2 with an overweight BMI). Two of the GDM cases were diagnosed based on the CGT results and prior history, while the remaining two completed the 3-hour OGTT.

**Table 1. Participant characteristics by pre-gravid body mass index group.**

| | All | Normalweight[3] (BMI < 24.9 kg/m$^2$) | Overweight[3] (BMI 25–29.9 kg/m$^2$) | Obese[3] (BMI >30.0 kg/m$^2$) | p-value[#] |
|---|---|---|---|---|---|
| **Number of participants (n%)** | 105 | 49 (46.7%) | 39 (37.1%) | 17 (16.2%) | - |
| **Race/Ethnicity** | | | | | |
| Caucasian | 58 (55.2%) | 35 (71.4%) | 17 (43.6%) | 6 (35.3%) | 0.07 |
| African American | 12 (11.4%) | 2 (4.1%) | 7 (17.9%) | 3 (17.6%) | |
| Hispanic/Latina | 29 (27.6%) | 8 (16.4%) | 13 (33.3%) | 8 (47.1%) | |
| Multiple Races | 1 (0.9%) | 1 (2.0%) | 0 (0%) | 0 (0%) | |
| Asian | 1 (0.9% | 1 (2.0% | 0 (0% | 0 (0% | |
| Other | 4 (3.8%) | 2 (4.1%) | 2 (5.2%) | 0 (0%) | |
| **Insurance status** | | | | | |
| Private Insurance | 53 (50.5%) | 33 (67.3%) | 16 (41.0%) | 4 (23.5%) | 0.01[**] |
| Medicaid | 23 (21.9%) | 7 (14.3%) | 11 (28.2%) | 5 (29.4%) | |
| Tricare | 4 (3.8%) | 3 (6.1%) | 1 (2.6%) | 0 (0.0%) | |
| Uninsured | 25 (23.8%) | 6 (12.2%) | 11 (28.2%) | 8 (47.1%) | |
| **Pre-gravid BMI (kg/m$^2$)[1]** | 26.4 +- 5.3 | 22.5 +- 2.6 | 27.2 +- 1.5 | 35.7 +- 4.1 | <0.00[***] |
| **Age (years)[1]** | 29.5 +- 5.0 | 29.4 +- 4.9 | 29.5 +- 5.6 | 29.5 +- 3.6 | 0.97 |
| **Antibiotic administration** | *Yes*: 24 (22.9%) *No*: 81 (77.1%) | *Yes*: 10 (20.4%) *No*: 39 (79.6%) | *Yes*: 8 (20.5%) *No*: 31 (79.5%) | *Yes*: 6 (35.3%) *No*: 11 (64.7%) | 0.41 |
| **Gravida[2]** | 2 (1–8) | 2 (1–6) | 2 (1–6) | 4 (2–8) | <0.00[***] |
| **Glucose Challenge Test (n = 103)** | | | | | |
| **Glucose value on the GCT (mg/dL)[1]** | 114.4 (27.3) | 110.8 (27.7) | 118.0 (28.2) | 116.9 (24.2) | 0.44 |
| **Women with greater than or equal to 140 mg/dL[3,4]** | | | | | 0.89 |
| Yes | 16 (15.2%) | 8 (16.3%) | 6 (16.7%) | 2 (11.8%) | |
| No | 87 (82.8%) | 41 (83.7%) | 31 (83.3%) | 15 (88.2%) | |
| **Oral Glucose Tolerance Test Results (n = 16)** | | | | | |
| **Fasting glucose value (mg/dL)[1]** | 83.06 (8.2) | 79.9 (5.3) | 82.4 (6.3) | 99.0 (5.7) | <0.00[***] |
| **Glucose value on the one-hour glucose tolerance test (mg/dL)[1]** | 142.9 (27.4) | 138.0 (26.4) | 137.6 (26.0) | 178.5 (9.2) | 0.14 |
| **Glucose value on the three-hour glucose tolerance test (mg/dL)[1]** | 88.9 (30.5) | 88.8 (28.7) | 81.6 (29.3) | 107.5 (54.4) | 0.63 |

[1]mean +- standard deviation

[2]median (range)

[3]n (%).

[4]Two participants did not have the GCT, one because of a previous diagnosis of diabetes mellitus and the other was lost to follow-up, both of these individuals were in overweight BMI group. Analysis of variance and follow-up Tukey's HSD test to determine pairwise differences for parametric continuous variables, Kruskal-Wallis rank sum test for non-parametric continuous variables, and chi-square testing and posthoc Fisher's exact test for categorical variables.

[*]p-value significance at 0.05.

[**]p-value significance at 0.01.

[***]p-value significance at <0.00.

## Differences in microbial taxa between impaired and normal glucose tolerance groups

Five taxa were significantly lower in women with IGT than those with NGT as shown in Fig 1. Taxa lower in women with impaired glucose tolerance were *Ruminococcacea* ($p$ = 0.01, log2FC = -1.7, FDR-adjusted $p$-value = 0.97), *Schaalia turicensis* ($p$ = 0.05, log2FC = -2.3, FDR-adjusted $p$-value = 0.97), *Oscillibacter* ($p$ = 0.03, log2FC = -1.7, FDR-adjusted $p$-value = 0.97),

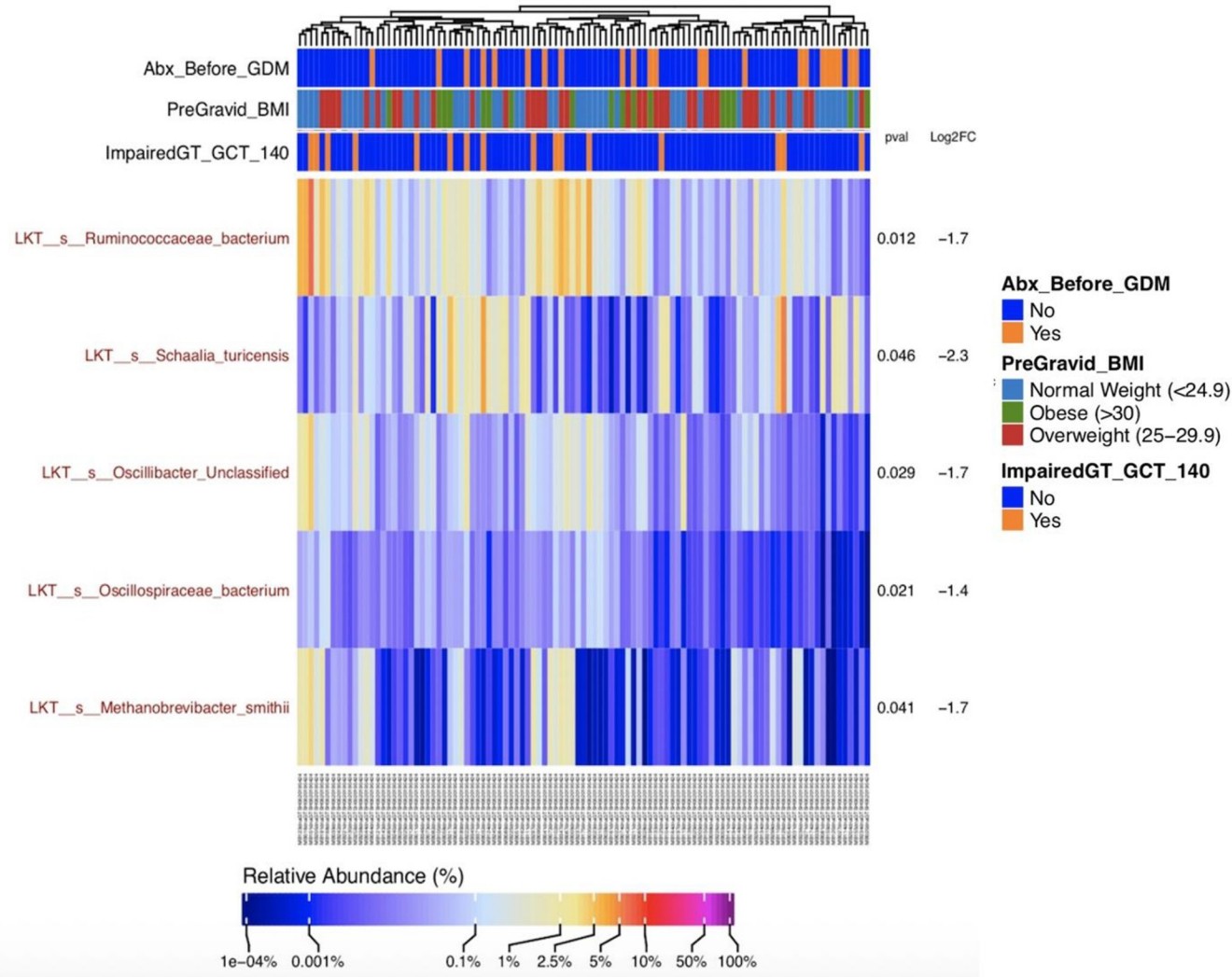

**Fig 1. A relative abundance heatmap of last known taxa comparison between individuals with impaired and normal glucose tolerance on the glucose challenge test.** Features included in this heatmap had to be present in at least 10% of biospecimens at a level of greater than 500 parts per million with a genome completeness sequencing greater than 0.1. Vertical columns represent each biospecimen included in the analysis (n = 105). Corresponding colors on the top three horizontal rows reflected clinical metadata described in the legend.

*Oscillospiraceae* ($p$ = 0.02, log2FC = -1.4, FDR-adjusted $p$-value = 0.97), and *Methanobrevibacter smithii* ($p$ = 0.04, log2FC = -1.7, FDR-adjusted $p$-value = 0.97). When we further compare those who have IGT by their pre-gravid BMI, five taxa are significantly different between the BMI groups, *Enterobacteriaceae* ($p$<0.00, FDR-adjusted $p$-value = 0.01), *Dialister micraerophilus* ($p$<0.00, FDR-adjusted $p$-value = 0.01), *Campylobacter ureolyticus* ($p$<0.00, FDR-adjusted $p$-value = 0.01), *Proteobacteria* ($p$<0.00, FDR-adjusted $p$-value = 0.03), *Streptococcus Unclassified* (species) ($p$<0.00, FDR-adjusted $p$-value = 0.04). Comparing participants who did and did not receive antibiotics prior to the glucose challenge test, there are no taxa that were significantly different, by both unadjusted and adjusted $p$ value, between participants who have IGT and NGT.

## Correlation between alpha diversity with impaired glucose tolerance

Table 2 shows the mean and standard deviation of four alpha diversity metrics (i.e., Shannon, Simpson, Inverse Simpson, and Chao1) by BMI group. There are no significant differences in

**Table 2. Mean and standard deviation for alpha diversity metrics between BMI groups.**

| | All | Normalweight[3] (BMI <24.9 kg/m$^2$) | Overweight[3] (BMI 25–29.9 kg/m$^2$) | Obese[3] (BMI >30.0 kg/m$^2$) | p-value |
|---|---|---|---|---|---|
| Simpson | 0.93 (0.06) | 0.93 (0.05) | 0.93 (0.04) | 0.90 (0.11) | 0.05 |
| Inverse Simpson | 19.62 (8.62) | 21.57 (8.59) | 18.57 (7.96) | 16.28 (9.28) | 0.07 |
| Shannon | 3.87 (0.49) | 3.96 (0.43) | 3.85 (0.42) | 3.63 (0.82) | 0.06 |
| Chao1 | 8731.81 (1101.40) | 8639.42 (1152.49) | 9006.13 (876.27) | 8366.50 (1322.91) | 0.11 |

alpha diversity by BMI. Further, across the entire sample of biospecimens, there was no significant correlation between blood glucose values and alpha diversity as shown in Table 3. There was a significant and positive correlation between fasting blood glucose and the 1-hour oral glucose tolerance test value ($r = .73$, $p < .01$). All four metrics including the Shannon ($p<0.00$), Simpson ($p<0.00$), Inverse Simpson ($p = 0.04$), and Chao1 ($p = 0.04$), showed a significant difference in alpha diversity with increased values in the IGT group (Fig 2).

## Association between microbial function and impaired glucose tolerance

Three GO terms had a positive log2-fold-change indicating an increase in the NGT group, including alpha amylase activity (GO:004556), oxidoreductase activity (GO:0016702), and autoinducer AI-2 transmembrane transport (GO:1905887). In the IGT group, five GO terms (55.6%) were molecular functions, including the catalysis of reactions related to aspartate dehydrogenase (GO:0033735), proteins at the C-terminal (GO:0004671), acetoin hydrogenase (GO:0019152) and daiaminobutyrate–pyruvate transaminase (GO:0047307). Further, four GO terms were biological functions, including methylation (GO:0006481), ectoine, and bacillithiol biosynthetic processes (GO:0019491, GO:0071793), and acetoin catabolic processes (GO:0045150). Fig 3 illustrates the relationship among the significant GO terms within the ectoine biosynthetic pathway. Table 4 shows more detailed information, including the definition, type of function, the direction of regulation, and the associated p-value for each identified term.

## Discussion

This study contributes needed examination on the association between microbial composition, diversity, and function of the maternal gastrointestinal microbiome with IGT on the GCT. No

**Table 3. Correlation between microbiome diversity metrics and maternal blood glucose values in the entire sample.**

| Variable | n | Mean | Standard deviation | 1 | 2 | 3 | 4 | 5 | 6 | 7 | 8 |
|---|---|---|---|---|---|---|---|---|---|---|---|
| 1) Glucose challenge test | 103 | 114.37 | 27.34 | - | | | | | | | |
| 2) Fasting | 16 | 83.06 | 8.23 | 0.08 | - | | | | | | |
| 3) 1-hour oral glucose tolerance test | 16 | 142.94 | 27.38 | 0.34 | 0.73** | - | | | | | |
| 4) 3-hour oral glucose tolerance test | 16 | 88.88 | 30.48 | -.10 | 0.22 | 0.43 | - | | | | |
| 5) Shannon | 103 | 3.87 | 0.49 | 0.09 | -.031 | 0.05 | -0.21 | - | | | |
| 6) Simpson | 103 | 0.93 | 0.06 | 0.04 | -0.32 | -0.14 | -0.47 | 0.89*** | - | | |
| 7) Inverse Simpson | 103 | 19.48 | 8.62 | 0.10 | -.032 | -0.09 | -0.45 | 0.85*** | 0.73*** | - | |
| 8) Chao1 | 103 | 8745.28 | 1090.47 | 0.07 | -0.22 | 0.19 | 0.11 | 0.51*** | 0.34*** | 0.33** | - |

*$p < 0.05$.

**$p <0.01$

***$p <0.00$.

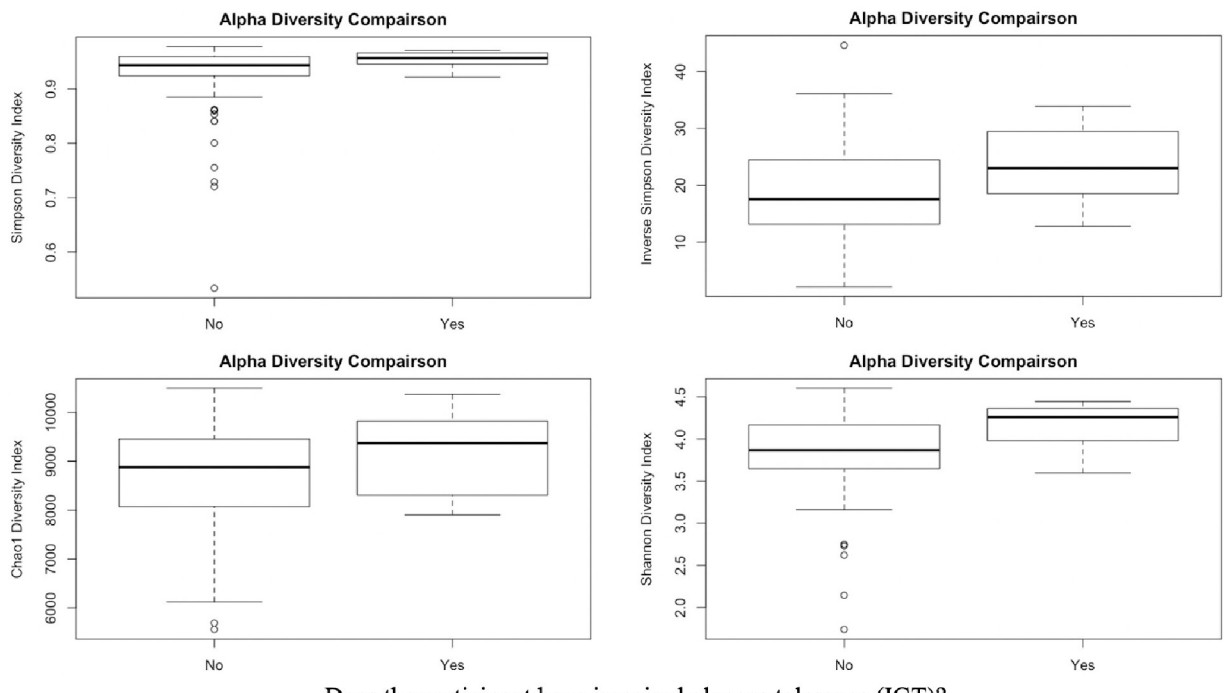

**Fig 2. Boxplots comparing alpha diversity metrics between impaired and normal glucose tolerance.**

currently published research has examined the potential of the microbiome to be a biological component of impaired glucose tolerance during pregnancy. Impaired glucose tolerance is a heterogeneous metabolic disorder, with the elevated blood glucose value establishing a metabolic phenotype similar to overt GDM [7], but is under-recognized in the clinical setting for recommendations on maternal-fetal surveillance and management.

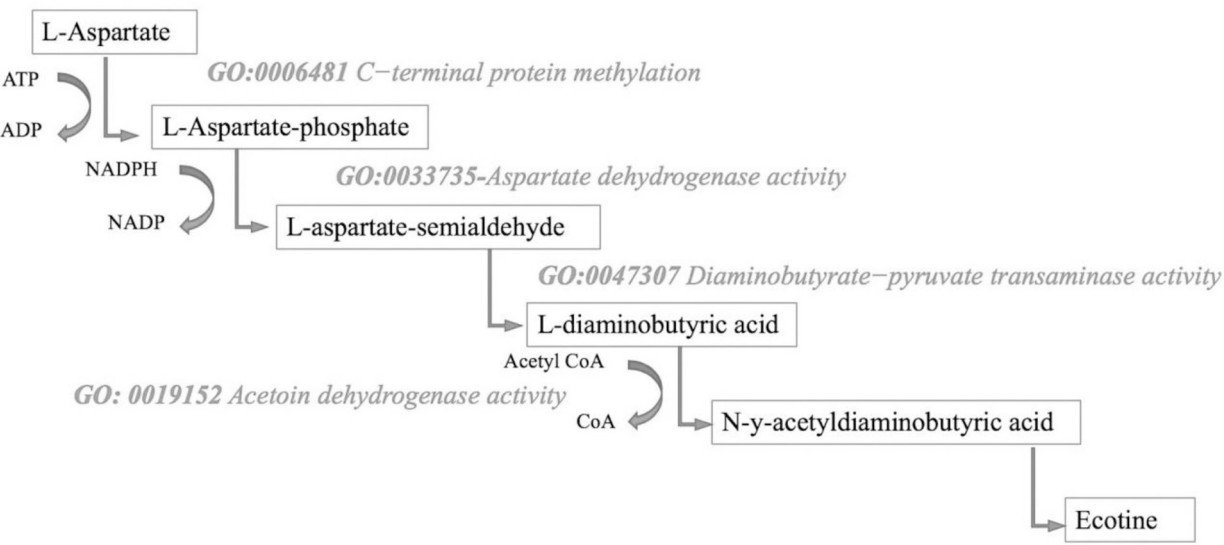

**Fig 3. GO terms increased in the IGT related to the ectoine biosynthetic pathway cascade.** From [30]. Copyright 2015, by He, Y. Z., Gong, J., Yu, H. Y., Tao, Y., Zhang, S., & Dong, Z. Y.

**Table 4. GO terms significant in between IGT and NGT groups.**

| Gene Ontology ID | Name | Type of Function^ | Gene Ontology Definition^ | Increased Expression Group | $p$-value# | Log2 Fold Change |
|---|---|---|---|---|---|---|
| GO:0004556 | Alpha amylase activity | Molecular | Catalysis of polysaccharides containing three or more alpha-linked D-glucose units. | NGT | 0.04 | 1.8 |
| GO:0016702 | Oxidoreductase activity | Molecular | Catalysis of an oxidation-reduction reaction. | NGT | 0.04 | 1.6 |
| GO:1905887 | Autoinducer AI−2 transmembrane transport | Biological | Transportation of autoinducer AI-2 across a membrane. | NGT | 0.05 | Inf |
| GO:0033735 | Aspartate dehydrogenase activity | Molecular | Catalysis of a reaction involving aspartate, water, and NAD(P) | IGT | 0.05 | 1.2 |
| GO:0004671 | Protein C−terminal S−isoprenylcysteine carboxyl O | Molecular | Catalysis of a reaction involving methionine and cysteine. | IGT | 0.02 | 1.1 |
| GO:0006481 | C−terminal protein methylation | Biological | The process of methylation of the C-terminal amino acid of a protein. | IGT | 0.02 | 1.1 |
| GO:0050118 | N−acetyldiaminopimelate deacetylase activity | Molecular | Catalysis of the reaction with diaminopimelate and water. | IGT | 0.02 | 1.1 |
| GO:0019491 | Ectoine biosynthetic process | Biological | The chemical pathways resulting in the formation of ectoine. | IGT | <0.00 | 2 |
| GO:0047307 | Diaminobutyrate−pyruvate transaminase activity | Molecular | The catalysis of the reaction of pyruvate and diaminobutyrate. | IGT | 0.02 | 1.7 |
| GO:0071793 | Bacillithiol biosynthetic process | Biological | The chemical pathways resulting in the formation of bacillithiol. | IGT | 0.01 | 2.2 |
| GO:0019152 | Acetoin dehydrogenase activity | Molecular | The catalysis of the reaction of acetoin and NAD. | IGT | 0.05 | 1.2 |
| GO:0045150 | Acetoin catabolic process | Biological | The chemical pathways resulting in the breakdown of acetoin. | IGT | 0.05 | 1.2 |

^Type of function, definition, and relevant genes were sourced from the AmiGO2 database [31,32].

#All $p$-values for significance were set at less than 0.05. GO term rows highlighted in gray are further mapped into the ectoine biosynthetic pathway provided in Fig 3.

In women identified as glucose intolerant by the GCT (n = 16), those with an obese BMI had significantly increased fasting blood glucose values ($p$<0.00). Fasting blood glucose values were inversely and weakly correlated with alpha diversity as measured by the Shannon diversity index ($r$ = .031, $p$ = 0.24). This value is not significant so limited interpretation should be made on its clinical relevance. Our non-significant finding of elevated fasting blood glucose with obesity substantiates a study by Harmon et al. [33], where women with obesity had higher fasting glucose profiles than normal-weight women despite being on a controlled diet since early pregnancy. Additionally, another research study showed that a higher fasting glucose in early pregnancy, in combination with an obese phenotype, places a woman at higher risk for developing complications for her health (i.e., cardiovascular disease, T2DM) and her baby's health (i.e., impaired fetal growth) [34]. Women with a prior diagnosis of GDM had increased prevalence of obesity and abnormal lipid profiles and impaired fasting glucose and IGT [35]. Therefore, fasting blood glucose levels may provide a more available biomarker for later pregnancy and delivery complications, though the relationship between microbial diversity remains to be elucidated as studies show mixed results between diversity indices and obese phenotypes [36]. Future research with higher sample sizes should consider the relationship between fasting blood glucose on the CGT with microbial diversity.

For the 16 participants with IGT on the GCT, there was a significant increase in alpha diversity across all measures. This may be the result of microbial compensation, regardless of BMI, where the maternal microbiome uptakes more of her microbial environment to promote symbiosis and normalized blood glucose values. This microbial compensation is able to withstand the pro-inflammatory processes and reduced microbial diversity of overt GDM but may

not be able to fully revert to an NGT microbial phenotype. Whether it is the change in the gastrointestinal environment of glucose intolerant women that drives changes in the microbes or the microbes that drive the changes in the host women remains to be elucidated.

The advantage of shotgun sequencing is the ability to compare the full genetic sequence of the microbiota between patients without regard to the assigned taxonomy of microbial species. This advantage is important as bacterial taxonomic signatures alone (such as those from 16S rRNA gene sequencing) are not always representative of function. Ecotine biosynthetic processes (GO:0019491) were increased in the gastrointestinal bacteria of women with IGT ($p < 0.001$). Ectoine is an amino acid widely used in organisms as a compatible solute to counterbalance osmotic environments in which proteins, membranes, or whole cells may be denatured. Aspartate dehydrogenase (GO:0033735) and Diaminobutyrate–pyruvate transaminase (GO:0047307) were also increased in the IGT group ($p = 0.05$ and $p = 0.02$). Both of these GO terms code for enzymes shared in the amino acid, aspartate, and are part of the ectoine biosynthetic pathway [30,37]. The only known substrate for aspartate-based phosphatase catalysis is the C-terminal domain of RNA polymerase II [16] and C-terminal based methylation reactions (GO:0006481, GO:0004671) were also up-regulated in the bacteria of women with IGT ($p = 0.02$, $p = 0.02$). These reactions require acetoin dehydrogenase activity (GO:0019152) which was again increased in the bacteria of the IGT group ($p = 0.05$).

Bacilitiol biosynthetic processes (GO:0071793) were also elevated in the IGT group ($p = 0.01$). These processes are important for detoxifying electrophiles [38]. Electrophiles are reactive proteins that in large doses can lead to toxicity, carcinogenesis, and cell death [39]. This finding continues to suggest the attempt of the maternal microbial gastrointestinal environment to overcome the challenges of reactivity and inflammation. The final GO process that was differentially increased in the bacteria of glucose intolerant women was acetoin catabolic processes (GO:0045150, $p = 0.05$). These processes are the major products of aerobic bacteria that breakdown large amounts of glucose without promoting acidification of their environment [40]. Hyperglycemia in humans can eventually lead to osmotic diuresis and acidosis. It was interesting that in our sample of glucose intolerant women, successful microbial residents of the gastrointestinal tract had upregulated mechanisms for combating cellular damage from dehydration and acidosis. Few bacterial species were differentially abundant in these women, perhaps due to the limited duration of pregnancy-induced hyperglycemia and the time necessary for specialized bacterial recruitment and colonization. Still, processes that are widely available to many species of bacteria were upregulated for survival in the gastrointestinal tracts of glucose intolerant women. Interestingly, one of the species that was differentially abundant in these women was *methanobrevibacter smithii* ($p = 0.041$), an archeon that specializes in scavenging hydrogen atoms and the fermentation products of bacteria specializing in the digestion of complex polysaccharides [41].

The Hyperglycemia and Adverse Pregnancy Outcome (HAPO) study, including 23,000 women in multiple centers throughout the world, found that the risk of adverse events to mother, fetus, and newborn increased with maternal glycemia even within normal ranges. Even 11 years later, women with only one abnormal glucose value had a higher risk of developing of type two diabetes mellitus and the complications associated with it [42]. Patterns of glucose values during the OGTT have been used to predict overall risk for future diagnosis of diabetes, cardiovascular disease, and all-cause mortality [43]. Reinforcing the role of big data in this clinical area, a cohort study of over 45,000 women Kaiser Permanente Northern California health system found that the risk of childhood obesity begins to increase at glycemic levels below the ability to diagnose GDM [44]. These studies, taken together, reiterate the need to address IGT even when there is no direct clinical deviation on the follow-up OGTT. Future researchers should consider utilizing medical record information, multi-omics analysis, and

clinical presentation together to help elucidate the mechanisms involved in the development of glucose intolerance during pregnancy and to stratify patients most at risk for both adverse pregnancy outcomes and future disease.

The reasons for glucose intolerance during pregnancy and why some women progress to gestationally induced diabetes are poorly understood. This study reveals that although the microorganisms are not taxonomically very different between glucose tolerant and intolerant women (with only five taxa out of over 13,000 identified as significantly different), the bacteria present in the gastrointestinal tracts of glucose intolerant women have upregulated systems to withstand stresses such as dehydration and acidosis. That the microbial community requires these adaptations indicates not only that the normal function of the microbiota such as nutrient production (GO:0004556), quorum sensing (GO:1905887), and free radical reduction (GO:0016702) are decreased in favor of survival mechanisms, but also that the cells of host gastrointestinal tract must be coping under stress as well. We know that there is a connection between low-grade inflammation and metabolic disease [15]. Cytokines such as TNFa and leptin that are already increased during pregnancy and, in the setting of obesity, may be further exacerbated by the bacterial overgrowth, dysbiosis, and increased vascular permeability seen during low level gastrointestinal inflammation. How this cycle begins and whether manipulation of the microbiota may be an avenue for therapeutic intervention is a suggested topic for future study.

There are several strengths and limitations to our study. Our primary strength is that we have metagenomically sequenced maternal gastrointestinal biospecimens to understand both the relevant taxa in participants with IGT and NGT but also the functional role that the microbiota play. However, our results are tamed by the lack of significance in our data that were significantly adjusted by false discovery rate. Only five taxa were identified significant by both the standard and FDR p-value. This is likely due to the overall low percentage of participants with IGT in our dataset making generalizability from these results difficult. Additionally, we understand that there are many unmeasured variables in our dataset that may impact microbial diversity and function such as dietary factors and other medication interactions. Though this was outside the scope of our study, we encourage future researchers to include dietary data in the primary hypotheses and analytic approach. Our preliminary results provide a foundation for future research in a larger sample focused on participants with IGT to further identify microbiota of relevance.

## Clinical implications

According to ACOG [29,45], current treatment for GDM includes nutrition therapy such as direct counseling, dietary changes that promote normoglycemia, preventing ketosis, and promotion of adequate gestational weight gain. ACOG acknowledges that patients with only one elevated value on the GGT may require additional surveillance (i.e., additional glucose testing, fetal monitoring, dietary counseling), but there are no broad policies for monitoring and treatment despite several studies implicating IGT as clinically salient, even into the postpartum state [46,47]. Outside of pregnant populations, IGT diagnosis occurs with blood glucose levels as low as 100 mg/dL during fasting [48]. IGT management includes moderate physical activity, controlled weight loss, and drug therapy to delay the onset of type 2 diabetes mellitus (T2DM) [48]. Future research and policy guidelines should consider how to adjust clinical surveillance of women with IGT on the GCT.

Interestingly, a dietary treatment study was conducted with 93 participants who were diagnosed with IGT or GDM to assess changes in insulin resistance during pregnancy (HOMA-IR) [49]. The dietary intervention did improve blood glucose values but did not sufficiently affect

insulin resistance or pertinent pregnancy outcomes [49]. It is possible that the microbiota plays a role in perpetrating the inflammation that leads to the insulin resistance at the core of this problem. More research should be done on the utility of interventions for preventing IGT, in addition to GDM, to help decrease the potential of women developing complications in pregnancy, delivery, and future diagnosis of T2DM.

## Conclusion

Our study highlights the gastrointestinal microbiome changes in women with IGT during pregnancy that require more follow up research to confirm. However, our results indicate a potential ability for women with IGT to support their blood sugar values with the gastrointestinal microbiota to stave off diagnosis of GDM in the short term, but that examination into longer term outcomes is imperative. Understanding the role of the microbiome in regulating glucose tolerance during pregnancy helps clinicians and researchers to understand the importance of IGT as a marker for adverse maternal and neonatal outcomes.

## Acknowledgments

The authors acknowledge Research Computing at The University of Virginia for providing computational resources and technical support that have contributed to the results reported within this publication (URL: https://rc.virginia.edu). Also, we would like to thank John McCulloch, Ph.D., Briana Cortez Chronister, Wuxing Yuan, and Laura Habermeyer for their roles in supporting this work.

## Author Contributions

**Conceptualization:** Caitlin Dreisbach, Jeanne Alhusen, Anna Maria Siega-Riz.

**Data curation:** Caitlin Dreisbach, Stephanie Prescott, Donald Dudley, Giorgio Trinchieri, Anna Maria Siega-Riz.

**Formal analysis:** Caitlin Dreisbach.

**Funding acquisition:** Caitlin Dreisbach, Jeanne Alhusen, Anna Maria Siega-Riz.

**Investigation:** Caitlin Dreisbach, Jeanne Alhusen, Giorgio Trinchieri, Anna Maria Siega-Riz.

**Methodology:** Caitlin Dreisbach, Jeanne Alhusen, Donald Dudley, Giorgio Trinchieri, Anna Maria Siega-Riz.

**Project administration:** Caitlin Dreisbach.

**Resources:** Stephanie Prescott, Jeanne Alhusen, Donald Dudley, Giorgio Trinchieri, Anna Maria Siega-Riz.

**Supervision:** Stephanie Prescott, Jeanne Alhusen, Donald Dudley, Anna Maria Siega-Riz.

**Validation:** Caitlin Dreisbach, Stephanie Prescott, Jeanne Alhusen.

**Visualization:** Caitlin Dreisbach.

**Writing – original draft:** Caitlin Dreisbach.

**Writing – review & editing:** Caitlin Dreisbach, Stephanie Prescott, Jeanne Alhusen, Donald Dudley, Giorgio Trinchieri, Anna Maria Siega-Riz.

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
