## [Decision Letter · Decision Letter 0]

21 Apr 2022

PONE-D-22-00832Association between microbial composition, diversity, and function of the maternal gastrointestinal microbiome with impaired glucose tolerance on the Glucose Challenge TestPLOS ONE

Dear Dr. Dreisbach,

Thank you for submitting your manuscript to PLOS ONE. After careful consideration, we feel that it has merit but does not fully meet PLOS ONE’s publication criteria as it currently stands. Therefore, we invite you to submit a revised version of the manuscript that addresses the points raised during the review process.

Thank you for submitting this manuscript. Firstly, I must apologise for the time it has taken to deal with the manuscript - this was caused by initial difficulties in recruiting sufficiently qualified reviewers. The manuscript has divided opinion among the reviewers. Reviewer #3 believes that the manuscript is fine as is. Reviewer #1 has some relatively minor points which need to be dealt with in any revised manuscript. However, reviewer #2 has some major concerns with the manuscript which need to be dealt with robustly for it to be accepted for publication. Of these I believe that the key points that need revision are the ones concerning the inclusion and exclusion of participants, pregnancy outcomes, and consideration of potential confounders. I also have one additional very minor point which I believe merits revision: in lines 47, 181, 189, 207 & Table 3 please change each instance of "(p=<0.00)" to "(p<0.01)" or use exact p-values.

We look forward to receiving your revised manuscript.

Kind regards,

Clive J Petry, PhD

Academic Editor

PLOS ONE

Journal Requirements:

Reviewers' comments:

Reviewer's Responses to Questions

**Comments to the Author**

1. Is the manuscript technically sound, and do the data support the conclusions?

Reviewer #1: Yes

Reviewer #2: No

Reviewer #3: Yes

2. Has the statistical analysis been performed appropriately and rigorously? 

Reviewer #1: Yes

Reviewer #2: No

Reviewer #3: Yes

3. Have the authors made all data underlying the findings in their manuscript fully available?

Reviewer #1: Yes

Reviewer #2: No

Reviewer #3: Yes

4. Is the manuscript presented in an intelligible fashion and written in standard English?

Reviewer #1: Yes

Reviewer #2: No

Reviewer #3: Yes

5. Review Comments to the Author

Reviewer #1: This is a well written and interesting cross-sectional study in an understudied population of women (pregnant women with GDM). This reviewer only has minor suggestions to the authors.

Line 35: Define “IGT” as an abbreviation here (i.e., “…impaired glucose tolerance (IGT)…”

Line 41: Define “NGT” as an abbreviation (e.g., “…and those without (normal glucose tolerance – NGT)…”)

Line 104-105: Please give a brief description of the heating/vortex protocols used to extract DNA from the swabs (e.g., temperatures used, length of incubation, etc.)

Lines 142-154: Was false discovery rate correction used? I would recommend FDR correction of your p-values because of the large number of comparisons done for this study; the FDR corrected results can be reported after the un-corrected results.

Lines 191-193: It is very difficult to see the difference in abundance of these bacteria between IGT and NGT because of how the data is presented. I would suggest using boxplots or log-fold-change bar plots to more clearly show the differences in abundance between IGT and NGT. Please also mention the results for antibiotic use and BMI category in-text and display those results in a boxplot/bar plot or remove them from the figure if they won’t be discussed.

Line 211-214: There are two “Table 2”s. The results in the first Table 2 (alpha diversity vs. BMI category) should be discussed in the text. The second Table 2 (glucose challenge vs alpha diversity) should be Table 3. The second table 2 is also missing p values and lacks in-text discussion of some of the results.

Lines 202-204: Results unclear. As stated, it sounds like Shannon/Inverse Simpson was negatively correlated with glucose challenge test, but the table shows a weakly positive correlation coefficient for both comparisons (0.09 and 0.10, respectively).

Lined 209-120: Remove the last sentence. Put “(Figure 2)” at the end of the previous sentence (i.e., …values in the impaired tolerance group (Figure 2).”). In the figure, please add a label to the x axis (e.g., “Does participant have IGT?”).

Reviewer #2: There are many studies about the intestinal flora of gestational diabetes mellitus. There are few studies on gestational impaired glucose tolerance. However, due to the mild severity of abnormal glucose tolerance during pregnancy, clinical research is of little significance

There are more statistical methods of intestinal flora that can be used

Lack of inclusion criteria and exclusion criteria for patients. No complete clinical data and pregnancy outcomes were provided, and no consideration was given to the situation that may affect the composition of intestinal flora, such as eating habits, application of antibiotics and exclusion of complications and complications during pregnancy

The discussion part directly cites the content of the result part. The reasoning and demonstration process part is lack of organization and supporting literature, which needs to be further summarized.

Reviewer #3: An interesting topic. Microbiota is very important for health. Glucose intolerance or impaired glucose metabolism alter the mintestinal microbiome, which in turn has consequences for the health of mother and child

6. PLOS authors have the option to publish the peer review history of their article (what does this mean?). If published, this will include your full peer review and any attached files.

Reviewer #1: No

Reviewer #2: No

Reviewer #3: No

---

## [Author Response · Author response to Decision Letter 0]

7 Jun 2022

Reviewer #1:

1) Thank you for this compliment and feedback. We appreciate your acknowledgement of the importance of this study in the context of women with impaired glucose tolerance. We believe that impaired glucose tolerance is an important biological clue for the health of a mother over the long-term. 

2) Thank you for this keen level of detail. This has been changed in the manuscript.

3) Thank you for this keen level of detail. This has been changed in the manuscript.

4) We have added in content to this section to further describe the Obstetric and Neonatal Outcomes Study (ONOS) biorepository protocols for preparing the samples for DNA extraction from the biospecimen swabs. We specifically addressed the heating and vortex procedures and associated temperatures for each step. Additionally, we have added more detail to the automated DNA extraction to include the exact kit identifier so readers can review the open protocol for DNA Isolation (e.g., the preparation of the sample, cell lysis, inhibitor removal, and isolation). The specific DNA isolation kit used in this study was the Qiagen MagAttract PowerMicrobiome DNA/RNA isolation kit (cat#27500-4-EP) and the protocol can be found in full detail on Qiagen’s website for secure download (https://www.qiagen.com/us/resources/download.aspx?id=f8bb41cc-02b3-42b4-8339-25be20c39174&lang=en).

5) We did use the false discovery rate (FDR) correction and have added those results to the manuscript with the unadjusted p-values. We did identify FDR-adjusted significant taxa between BMI group for those who have impaired glucose tolerance (n=16). This content has been added to the results section. This was not commented on previously because the focus was the comparison between IGT and NGT, not pre-gravid BMI group in the IGT group (though we do believe it is relevant in this manuscript). Unfortunately, after correcting for multiple comparisons in the taxa difference between participants with IGT and participants with NGT across the entire sample (n=105), the significance is no longer present with FDR correction which is a limitation of this study. We had also added content in the discussion regarding this limitation. 

6) Thank you for this comment on Figure 1. We had difficulty identifying a type of visualization that would provide the amount of detail that the Complex Heatmap can display. For this reason, we have made adjustments to the figure to optimize the view rather than adjusting to a box plot. We do provide boxplots for alpha diversity comparisons later in the manuscript. Additionally, we appreciate your identification on the lack of in-text content on antibiotic administration. We have added this content in the results and kept it in the figure.

7) Thank you for this detail. We have changed the tables to the proper order and headings. We have also added more in-text discussion to better highlight the results from each of these tables. P-values were added to Table 3 showing the correlations between diversity metrics and maternal blood glucose values at each of the timepoints. Further, we’ve added the number of participants that were included at each of the glucose assessment timepoints.

8) Thank you for this comment. We have adapted both the correlation table (Table 3) and the content in the results section. We hope this better synthesizes the tables and content together. 

9) These edits have been made in the manuscript. Thank you for the comments that add to the clarity of the figures.

Reviewer #2

1) We appreciate your comment on the current clinical utility and significance of impaired glucose tolerance during pregnancy. There are several studies that suggest that the true impact of impaired glucose tolerance during pregnancy is that impaired glucose tolerance could influence the development of type 2 diabetes in the future (1,2). We hope that our manuscript helps to identify areas of future research that can be leveraged for a more rigorous multi-omics study that includes other potential confounding variables such as dietary intake and the assessment of insulin resistance.

2) We completely agree with this statement as microbiome and genomic analytic approaches have flourished in recent years. We acknowledge that there are many statistical approaches that we could have used for the purposes of this study. Further, there are many bioinformatic approaches to assemble and identify the taxonomic variables. We chose to use metagenomic sequencing to capture the taxa and functional differences between participants with impaired glucose tolerance and those with normal glucose tolerance. By focusing on metagenomic sequencing as our genomic approach, we had to narrow our potential analytic methods. We used the JAMS pipeline to efficiently assemble both our taxa and functional terms. This pipeline and associated statistics were recently implemented in two papers published in Science by our collaborating team (3,4).

3) We have added a section in the methods to outline the inclusion and exclusion criteria for the parent study, ONOS, and for this study on impaired glucose tolerance. To comply with proper reporting guidelines and to improve the current state of the manuscript, we used the STrengthening the Reporting of OBservational studies in Epidemiology (STROBE) checklist and the Patient Flow diagram to make it clearer as to how we arrived at our final sample size for each analysis in the study. The STROBE Checklist and Patient Flow Diagram are uploaded in the revision as supplementary files. We also provided a better description of the relevant clinical data that we pulled from the medical record system to give a more detailed outline of the data that we considered in the analysis. These variables were chosen because they may influence the composition or function of the gastrointestinal flora or place an individual at high risk of developing impaired glucose tolerance or gestational diabetes. As you can appreciate, we were not able to comprehensively asses every variable that may have an impact on the composition of the gastrointestinal microbiome. We completely agree that dietary patterns and eating habits play a role in composition and future work is necessary to corroborate currently published literature. As such, we added additional content to the discussion to address this as a limitation and provide a statement on leveraging unmeasured variables in our data as future research.

4) Thank you for this comment. We have adjusted the in-text for both the results and the discussion that we hope to better align the content and contextualize the findings. Further, we added a component to the discussion that addresses the strengths and limitations of this study that helps to tame the over-interpretation of the results.

Reviewer #3

1) Thank you for your interest in our study. We appreciate that you acknowledge the importance of glucose metabolism and the maternal gastrointestinal microbiome to impact the health of mothers and babies in the long-term. We have an associated study using the same biospecimens that links the maternal gastrointestinal microbiome to the newborn outcome of birthweight adjusted for gestational age. The manuscript is currently under revision at Pediatric Research. We hope that the literature continues to grow in this area.

---

## [Editor Report · Decision Letter 1]

28 Jun 2022

Association between microbial composition, diversity, and function of the maternal gastrointestinal microbiome with impaired glucose tolerance on the Glucose Tolerance Test

PONE-D-22-00832R1

Dear Dr. Dreisbach,

We’re pleased to inform you that your manuscript has been judged scientifically suitable for publication and will be formally accepted for publication once it meets all outstanding technical requirements.

Kind regards,

Clive J. Petry, PhD

Academic Editor

PLOS ONE
---

## [Editor Report · Acceptance letter]

1 Nov 2022

PONE-D-22-00832R1 

Association between microbial composition, diversity, and function of the maternal gastrointestinal microbiome with impaired glucose tolerance on the Glucose Challenge Test 

Dear Dr. Dreisbach:

I'm pleased to inform you that your manuscript has been deemed suitable for publication in PLOS ONE. Congratulations! Your manuscript is now with our production department. 

Kind regards, 

on behalf of

Dr. Clive J. Petry 

Academic Editor

PLOS ONE